# Sediment Classification of Acoustic Backscatter Image Based on Stacked Denoising Autoencoder and Modified Extreme Learning Machine

**Ping Zhou [1], Gang Chen [1,2,\*], Mingwei Wang [3], Jifa Chen [1]**  **and Yizhe Li [1]**

[1] College of Marine Science and Technology, China University of Geosciences, Wuhan 430074, China; pingzhou@cug.edu.cn (P.Z.); chenjifa@cug.edu.cn (J.C.); yizheli@cug.edu.cn (Y.L.)

[2] Hubei Key Laboratory of Marine Geological Resources, China University of Geosciences, Wuhan 430074, China

[3] Institute of Geological Survey, China University of Geosciences, Wuhan 430074, China; wangmingwei@cug.edu.cn

[\*] Correspondence: ddwhcg@cug.edu.cn; Tel.: +86-138-0713-4417

**Abstract:** Acoustic backscatter data are widely applied to study the distribution characteristics of seabed sediments. However, the ghosting and mosaic errors in backscatter images lead to interference information being introduced into the feature extraction process, which is conducted with a convolutional neural network or auto encoder. In addition, the performance of the existing classifiers is limited by such incorrect information, meaning it is difficult to achieve fine classification in survey areas. Therefore, we propose a sediment classification method based on the acoustic backscatter image by combining a stacked denoising auto encoder (SDAE) and a modified extreme learning machine (MELM). The SDAE is used to extract the deep-seated sediment features, so that the training network can automatically learn to remove the residual errors from the original image. The MELM model, which integrates weighted estimation, a Parzen window and particle swarm optimization, is applied to weaken the interference of mislabeled samples on the training network and to optimize the random expression of input layer parameters. The experimental results show that the SDAE-MELM method greatly reduces mutual interference between sediment types, while the sediment boundaries are clear and continuous. The reliability and robustness of the proposed method are better than with other approaches, as assessed by the overall classification effect and comprehensive indexes.

**Keywords:** sediment classification; acoustic backscatter image; stacked denoising auto encoder; extreme learning machine; Parzen window; particle swarm optimization

## 1. Introduction

The different types of seabed sediments provide important reference information for scientific research via seabed geological surveys, marine engineering construction, marine space planning, and benthic communities [1–4]. Traditional snapshot sampling and underwater photography are inefficient and costly for sediment classification [5]. On the contrary, it is feasible to utilize an intensity image of acoustic backscattering to judge the type of underwater sediments present, based on a certain number of samples. The usual approach is to conduct feature extraction and obtain the category labels via a classifier, which have high measurement efficiency and wide adaptability. However, backscatter images generated by acoustic sonar data are still prone to producing the phenomenon of drag, astigmatism and scattered speckles due to the influences of hull attitude, seabed topography fluctuation and ocean reverberation [6]. In addition, uncorrectable seafloor tracking processing areas exist in the middle of

the side-scan image; these areas are caused by the measurement mechanism. Therefore, this poses a daunting challenge for the classification of seabed sediments based on the backscatter image.

In general, numerous researchers have studied the aspects of feature extraction and classifier design. Feature extraction involves extracting objective indicators or in-depth hidden information from the backscatter image; these extracted indicators and information are then accurately applied to characterize the sediment type and properties. Commonly used feature indexes include: basic probability distribution parameters [7], the terrain slope [8], the gray-level co-occurrence matrix (GLCM) [9], the power spectrum characteristics [10], the angular response curves [11], the Gabor texture and the seabed roughness [12]. However, these feature indexes are easily disturbed by residual errors in the original data. In addition, deep learning, having strong learning ability, has attracted considerable interest in the terms of feature extraction. Among the deep learning approaches, a convolutional neural network (CNN) was adopted in a previous paper to extract sample features from large-scale side scan sonar images [13]. However, it was difficult to capture the long-distance features because of the perceptual field of view. On the other hand, a stacked auto encoder (SAE) [14] was applied in another paper to extract the hidden features from hyperspectral images using an unsupervised approach, but the risk of overfitting with the SAE was affected by the model complexity, the number of training samples, and data noise. Therefore, a stacked denoising auto encoder (SDAE) was used to enhance the robustness and generalization ability of the SAE model. At present, the SDAE model is widely used in data reduction and feature extraction, especially for extracting robust features from remote sensing images [15–17]. Similarly, acoustic backscatter images have the same scattering characteristics as remote images from the synthetic aperture radar, so it is necessary to extract the robust features for subsequent sediment classification.

The selection of the classifier is also important after the extraction of sediment features. Commonly unsupervised classification techniques include K-means [18], self-organizing feature maps [19] and iterative self-organizing data analysis (ISODATA) [20]. These methods are greatly influenced by noise data, resulting in the rapid reduction of classification accuracy. Some post-processing methods are needed to optimize the classification effect when drawing the preliminary seabed sediment map, such as Bayesian technology [21]. Additionally, the supervised classification techniques of seabed sediments include neural network (NN) [22], multilayer perceptron [23], random forest (RF) [24] and support vector machine (SVM) [25] approaches. However, the sensitivity of the key parameters and the robustness of these supervised classifiers are usually restricted. Ojha and Maiti applied Bayesian and NN to distinguish sediment boundaries in the Bering Sea slope area [26]. This method effectively improves the robustness of the algorithm when there is red noise in the data. Meanwhile, swarm intelligence algorithms could be used to optimize parameters of classifiers. For example, a NN with a genetic algorithm (GA) [27] and an SVM with particle swarm optimization (PSO) [28] were used to classify the sediments in offshore areas and islands, achieving better classification results. However, the classification sensitivity of these techniques needs to be improved in order to distinguish more types of sediments and the small differences in category characteristics.

Compared with the above classifiers, the network framework of the extreme learning machine (ELM) is capable of obtaining higher scalability and faster learning efficiency. In recent years, the ELM framework has been widely applied in sample prediction [29], regression analysis [30] and image classification [31]. In addition, many scholars have improved solutions to the problems related to the application of the basic ELM model. Ren et al. [32] used simulated annealing (SA), GAs and PSO to optimize the ELM model parameters in order to construct cognitive models of Chinese black tea ranks, whereby the stability of the ELM model was improved and much of the redundant information was removed. Man et al. [33] adopted regularization theory to optimize the random input weights of the ELM model, so as to achieve the minimum risk value and enhance the anti-interference of noise data. He et al. [34] used a Parzen window to solve the imbalanced classification problem of ELM. However, this integrated model only solved the double-classification problem, while the weight estimation of the samples during multi-classification was more complicated. In summary, it is necessary to optimize the

ELM model for image classification in terms of the randomness of the input layer parameters, the noise in the training samples and sample imbalance.

Therefore, a sediment classification method based on acoustic backscatter image is proposed, uniting the SDAE and the modified extreme learning machine (MELM) model for feature extraction and category prediction in the survey area (Figure 1). The MELM model includes two modules: a robust extreme learning machine (RELM) and an intelligent optimization process for PSO. In this work, we make the following progress: (1) SDAE is applied to extract the deep-level features of sediments, so that the training network automatically learns how to remove the residual errors and noise from the backscatter image; (2) RELM introduces weighted estimation and a Parzen window to improve the stability of unbalanced samples and weaken the interference of mislabeled samples on the network; (3) In order to obtain the optimal expression of network parameters, PSO is used to solve the randomness problem of the input layer parameters.

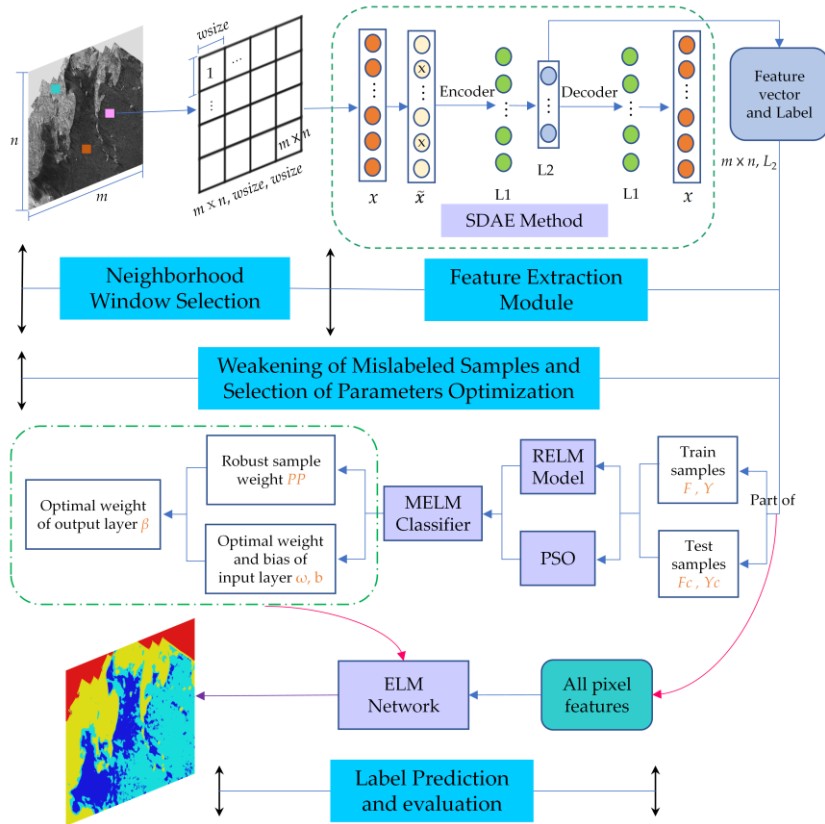

**Figure 1.** Flowchart of sediment classification based on the acoustic backscatter image.

The remainder of this paper is described as follows. Section 2 illustrates the proposed sediment classification principles and processes in detail. Experimental results and analysis are outlined in Section 3. Certain relative discussions are provided in Section 4. Finally, the conclusions are presented in Section 5.

## 2. The Proposed Sediment Classification Technique

### 2.1. Overall Framework

The classification process for seabed sediments involves determining the category of each pixel in the backscattered sonar image. In this paper, an SDAE feature extraction and MELM classifier are combined for sediment classification (Figure 1). The proposed technology mainly includes: (1) a neighborhood window selection of a pixel sub image; (2) feature extraction by SDAE; (3) weakening of

mislabeled samples and parameter optimization of the MELM classifier; and (4) prediction of sediment types and quality evaluation.

Firstly, the size of the neighborhood window is set; then the neighborhood sub-image of each pixel is obtained via sliding window sampling. Secondly, an unsupervised SDAE approach is applied to extract the deep-seated feature information from their sub-images. The feature information and labels corresponding to the training and testing samples are then selected and sent to the MELM classifier. The interference effect of the mislabeled samples on the network is weakened by weighted estimation and the Parzen window, and optimal input parameters of the network are selected by PSO. Finally, the feature vector and label information of the testing samples are input into the training network to obtain the category of prediction labels. Many objective metrics are applied to quantitatively evaluate the sub module of the classification method. Moreover, the feature information extracted from all pixels is input into the ELM network to obtain the sediment prediction map of the entire area, while the performance of the proposed method is analyzed from the perspective of a subjective visual effect.

### 2.2. SDAE Method for Feature Extraction

The traditional SAE approach is used for multi-dimensional feature extraction, which is limited by the data quality and model complexity, and inevitably results in the over-fitting phenomenon. Therefore, Vincent et al. proposed a de-noising auto encoder (DAE) to make the input signal more robust, and to improve the generalization ability of the network [35]. In the DAE process of feature extraction, the training network learns to remove the noise and residual errors from the original image using a zero masked fraction. In general, multiple DAEs are stacked to form a deep network model named the SDAE, so as to extract the deep-seated abstract features of image information [36].

Figure 2 shows a two-layer hidden SDAE network. Firstly, neighborhood pixels are selected to represent the sub-image of the sample point center. Secondly, the sub-image is expanded into a column as the input signal $x$, and the zero masked fraction is added with a certain probability to form a damaged signal $\widetilde{x}$. Signal $\widetilde{x}$ is encoded by linear mapping and the nonlinear activation function.

$$e = fg(pw_1 \cdot \widetilde{x} + pb_1) \tag{1}$$

where $pw_1$ and $pb_1$ are the weight and bias between the coding layer and input layer, respectively and $fg\left(\cdot\right)$ is the node activation function, such as a sigmoid function.

The reconstructed signal $rx$ is obtained by deciphering the coding features:

$$rx = fg(pw_2 \cdot e + pb_2) \tag{2}$$

where $pw_2$ and $pb_2$ are the weight and bias between the decoding layer and coding layer.

Finally, the reconstruction error $re$ between the signal $rx$ and $x$ is defined as follows:

$$re = \operatorname{argmin} \sum_{i=1} \|rx_i - x_i\|^2 \tag{3}$$

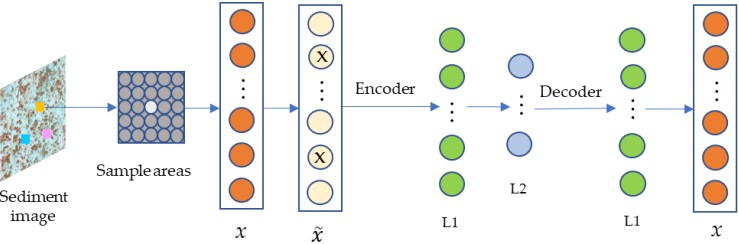

**Figure 2.** Feature extraction process of the stacked denoising auto encoder (SDAE).

The reconstruction error is backpropagated iteratively to optimize the structural parameters of the network. In this way, the encoder is able to extract deep-seated abstract features, making the learned feature data more robust. Therefore, the implicit information for the lowest layer L2 is utilized to represent the feature vector of the sample point, which serves as the input value of the subsequent classifier to achieve label classification of image pixels.

### 2.3. ELM and Its Modified Model

#### 2.3.1. Basic ELM Model

The ELM model is a feedforward neural network with a single hidden layer. It has been proven that the ELM possesses the capability for global approximation as a neural network [37]. The strategy involves adopting the least square theory to reverse calculate the output weight of the hidden layer by randomly setting the weights and bias of the input hidden layer. The principles of this method are shown in Figure 3.

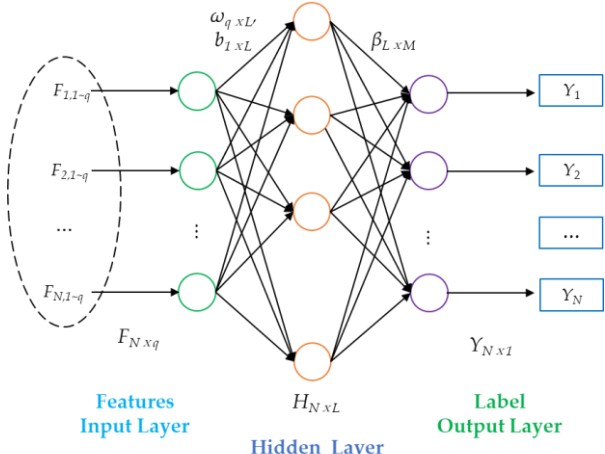

**Figure 3.** Network model of the extreme learning machine.

A feature dataset and corresponding labels are selected as the training samples $\{(F_j, Y_j)\}_{j=1}^{N} \subset R^N \times R^q$, giving a total of $N$ samples. The input feature dimension is $q$, and the maximum category number of label $Y_j$ is $m$. Suppose the number of hidden layers is $L$. The activation function is $g(f)$, and the weight and bias parameters $(w_i, b_i)$ of input hidden layer are randomly selected $(i = 1, \dots L)$. Therefore, the relationship between the feature set $F_j$ $(j = 1, \dots, N)$ and the output label $Y_j$ is:

$$\sum_{i=1}^{L} g\left(w_i \cdot F_j + b_i\right) \cdot \beta_i = Y_j, \quad j = 1 \ \cdots N \tag{4}$$

where $w_i = [w_{i,1}, w_{i,2}, \dots, w_{i,q}]^T$ is the input weight of $i$-th hidden layer and $b_i$ is its input bias. The activation kernel function $g(f)$ is usually represented by the sigmoid function, radial basis function, and polynomial function [38].

Importing $N$ feature samples into a matrix form, the ELM network is expressed as:

$$H\beta = Y \tag{5}$$

$$H = \begin{bmatrix} h(f_1) \\ \vdots \\ h(f_N) \end{bmatrix} = \begin{bmatrix} g(w_1 \cdot f_1 + b_1) & \cdots & g[w_L \cdot f_1 + b_L] \\ \vdots & \ddots & \vdots \\ g(w_1 \cdot f_N + b_1) & \cdots & g(w_L \cdot f_N + b_L) \end{bmatrix}_{N \times L}, \ \beta = \begin{bmatrix} \beta_1 \\ \vdots \\ \beta_L \end{bmatrix}_{L \times M}, \ Y = \begin{bmatrix} y_1 \\ \vdots \\ y_N \end{bmatrix}_{N \times M} \tag{6}$$

where $H$ and $\beta$ are the output matrix and weight of the hidden layer respectively, and $Y$ is the index value of the label category number. The number of the label category is converted into a positional index value in the sample classification problem. For example, if the maximum category is $M$, the third position of category number 3 is 1, and the others are 0, $y_3 = [0, 0, 1, \dots 0]_{1 \times M}$.

Integrating the least squares criterion with the minimum output weight norm, a regularization coefficient is introduced to ensure that when the output matrix of hidden layer $H$ has a rank deficiency problem, its generalized Moore inverse can be obtained correctly. Finally, the weight $\beta$ of the output layer is obtained by an inverse calculation.

$$\min: \quad \frac{1}{2}\|\beta\|^2 + \frac{1}{2}C\sum_{i=1}^{N}\|\varepsilon_i\|^2 \tag{7}$$

$$\beta = \left(\frac{I}{C} + H^T H\right)^{-1} H^T Y \tag{8}$$

where $C$ is the regularization coefficient, $I$ is the identity matrix, and $\varepsilon_i$ is the training sample error.

All image feature sets $Fp$ are brought into the trained ELM network model, meaning it is easy to quickly obtain the prediction label, $Yp = Hp \cdot \beta$. In addition, test samples are applied to analyze the accuracy of the ELM model.

### 2.3.2. MELM Model

The MELM model includes the theory of the RELM model and the PSO process with optimal parameter selection.

### Theory of RELM Model

Regional block outliers are usually found in backscatter intensity images, which inevitably lead to incorrect information in the selection of training samples [39]. The basic ELM model introduces error information to guide the calculation of the network parameters, which seriously interfere with the classification of seabed sediments. In addition, an unbalanced quantity occurs in the sediment category survey and the equal weight observation in the basic ELM model reduces the classification accuracy of the minimal sample class. Therefore, a Parzen window [34] and weighted estimation [40] are introduced in the ELM model (abbreviated as RELM) to weaken the influence of mislabeled training samples and unbalanced data on the network. The framework process for the RELM model is shown in Figure 4.

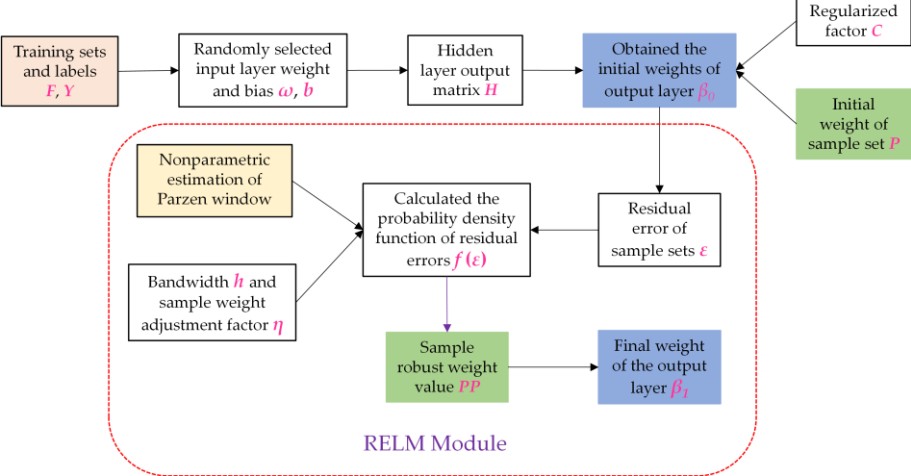

**Figure 4.** Framework flow of the robust extreme learning machine (RELM) Model.

The weight $\beta$ of the output layer in the robust ELM model becomes:

$$\beta = \left(\frac{I}{C} + H^T P H\right)^{-1} H^T P Y \tag{9}$$

where $P$ is the weight matrix of the sample observation, $P = \text{diag}\{p_1, p_2, \ldots, p_N\}$, which is regarded as the identity matrix in the basic ELM model.

We consider a case where there are a few samples with label errors in the selected training feature set. Firstly, the training sample set is introduced into the ELM network to obtain the initial weight $\beta_0$ of the output layer. Secondly, the initial residual error $\varepsilon$ is obtained, $\varepsilon = H \cdot \beta_0 - Y$. The probability density function $f(\varepsilon)$ of the sample residual is estimated by using the residual matrix $\varepsilon$. The distribution of $f(\varepsilon)$ occurs via the estimation without prior probability information. The Parzen window is used for nonparametric estimation in the case of limited samples. The window function $\psi(\kappa)$ is a normal function.

$$f(\varepsilon) = \frac{1}{N}\sum_{i=1}^{N}\frac{1}{h_N}\psi\left(\frac{\varepsilon - \varepsilon_i}{h_N}\right) \tag{10}$$

$$\psi(\kappa) = \frac{1}{\sqrt{2\pi}}\exp\left(-0.5 \cdot \kappa^T \cdot \kappa\right) \tag{11}$$

where $N$ is the total number of training samples, and $h$ is the window width. The term $h_N$ is the adjustment coefficient of the sample window width, $h_N = h/\sqrt{N}$, $\kappa = (\varepsilon - \varepsilon_i)/h_N$.

For sample $j$, the probability density function $f(\varepsilon_j)$ of its residual error is calculated by Equations (10) and (11). Therefore, the weight $pp_j$, after robust estimation is adjusted as follows:

$$pp_j = p_j \cdot \left[1 + \eta\frac{f(\varepsilon_j) - \min(f(\varepsilon))}{\max(f(\varepsilon)) - \min(f(\varepsilon))}\right] \tag{12}$$

where $p_j$ is the original weight and $\eta$ is the weight adjustment factor, which ensures the weight distribution in the range of $[1, 1 + \eta]$.

A new weight matrix of the robust estimation $PP$ is acquired, which is introduced into Equation (9) to obtain the final output layer weight $\beta_1$ of the RELM model, $PP = \text{diag}\{pp_1, pp_2, \ldots, pp_N\}$. The weight value of the reliable samples is increased, which is beneficial in improving the overall robustness of the network and in accurately classifying the label categories of feature samples.

RELM Model and PSO Combined into MELM Classifier

The random selection of the input weights and deviations causes the ELM model to generate overfitting and instability problems [41], requiring a swarm intelligence algorithm to optimize its parameters. Common swarm intelligence algorithms include the ant colony algorithm [42], GA [27], and PSO [28]. In this paper, the PSO and RELM model are combined into the MELM classifier to optimize the weights and biases of the input layer (Figure 5). The fitness values are expressed by the output errors of testing samples.

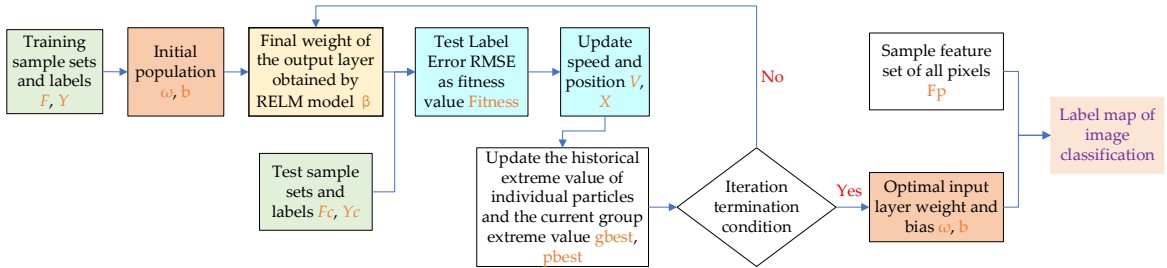

**Figure 5.** Intelligent optimization process of modified extreme learning machine (MELM) model.

Figure 5 shows the process of the MELM model. Firstly, the weight $w$ and bias $b$ of the input layer represent the characteristics of individual particles using the position and speed information. Then, each particle is calculated using the RELM model to obtain the corresponding robust weight of the output layer. Similarly, the root mean square error (RMSE) of the testing sample is employed to characterize the fitness of each particle. Finally, the particles move within a certain search space and share information with each other. The current position and speed information are updated by tracking individual historical optimal extremum and population extremum. The optimal particle individual is obtained until the whole population reaches the optimal state. The MELM model integrates the historical global optimization and memory functions of PSO and the robust learning ability of the RELM.

### 2.4. Evaluation Indexes of the Classification Model

In general, the classification of seabed sediment adopts subjective visual effect and objective qualitative indexes to evaluate the classification performance of each model. Subjective vision can clearly reflect the misclassification area and category interference factors. In addition, this paper adopts the verification indicators of the test set for analysis, mainly including the overall accuracy (*OA*), category accuracy (*CA$_i$*), *kappa* coefficient and *RMSE* values of the label prediction [11,25,43]. Their mathematical expressions are shown in Table 1.

**Table 1.** Mathematical descriptions of objective indexes for sediment classification.

| Objective Indexes | Mathematical Formulation |
|---|---|
| Overall Accuracy | $OA = \left( \sum\limits_{i=1}^{M} Num_{ii} \Big/ \sum\limits_{i=1}^{M} \sum\limits_{j=1}^{M} Num_{ij} \right) \times 100\%$ |
| Category Accuracy | $CA_i = \left( Num_{ii} \Big/ \sum\limits_{i=1}^{M} \sum\limits_{j=1}^{M} Num_{ij} \right) \times 100\%$ |
| *Kappa* Coefficient | $Kappa = \frac{OA - OE}{1 - OE}, OE = \left\{ \sum\limits_{k=1}^{M} \left( \sum\limits_{i=1}^{M} Num_i \times \sum\limits_{j=1}^{M} Num_j \right)_k \right\} \Big/ \left( \sum\limits_{i=1}^{M} \sum\limits_{j=1}^{M} Num_{ij} \right)^2$ |
| *RMSE* of label prediction | $RMSE = \sqrt{\frac{1}{Nc} \sum\limits_{i=1}^{Nc} \sum\limits_{j=1}^{M} \left( yc_{ij} - y_{ij} \right)^2}$ |

Where $M$ is the number of categories. $Num_{ij}$ represents the number of the $i$-th sample predicted to be in the $j$-th class. $Num_{ii}$ represents the number of the $i$-th sample predicted to be in the $i$-th class. $Num_i$ is defined as the number of real samples in the $i$-th class. $Num_j$ represents the number of samples predicted in the $j$-th class. $Nc$ is the total number of test samples. The terms $y_{ij}$ and $yc_{ij}$ are the position index value and prediction value of the test label, respectively.

### 3. Results and Analysis

All the codes were based on MATLAB 2019a and run on the Windows 10 platform. Two datasets were designed to analyze the influence of the feature extraction model and classifier performance on sediment classification. The superiority of the proposed method and modules is reflected in subjective vision and objective evaluation index values, as shown in this section.

### 3.1. Data Description and Parameter Settings

Two datasets were applied to verify the effectiveness and reliability of the adopted feature extraction method and the improved classifier. Dataset 1 was derived from a collaborative effort result between the United States Geological Survey (USGS) and the Massachusetts Office of Coastal Zone Management (CZM). The project provided a high-resolution geophysical data set covering the entire seafloor, supplemented by sediment samples and seabed photographs [44]. In this paper, we selected the post-processing 1×1 m high-resolution backscattered intensity image, and the Z1 area was chosen

to analyze the fine types of sediment. The sediment types in the experimental areas were divided into: fine silt, median silt, fine sand, median sand, granule and undetected area (Figure 6a). The distribution of seabed sediments in the Z1 area is relatively complicated. In addition, there were many residual errors in the Z1 area after the preprocessing process, inevitably requiring a small amount of false sample labels to be manually calibrated. In order to maintain the equilibrium of samples, we selected the sediment classes with fewer categories as much as possible, so as to enhance the number of those categories and weaken the randomness of the ELM family model. The number of training and testing samples is shown in Table 2.

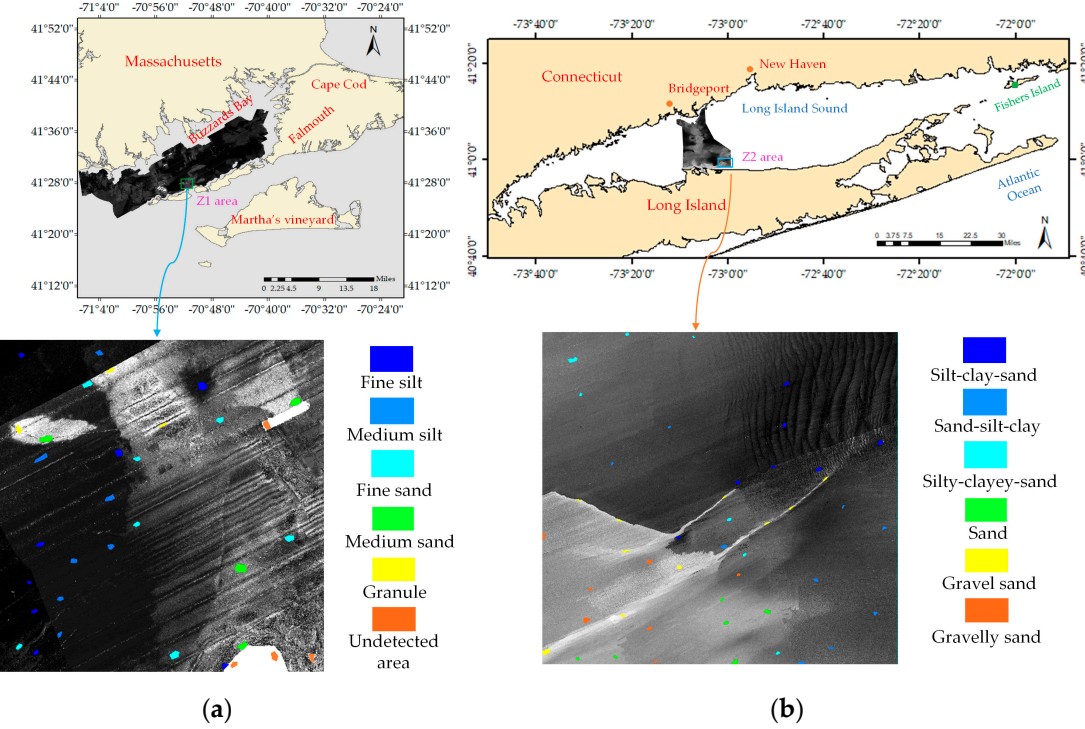

(**a**)          (**b**)

**Figure 6.** Locations of study areas 1 and 2 and acoustic backscatter images. (**a**) Sediment image of the Z1 area, and label distribution of the training samples from Dataset 1. (**b**) Sediment image of the Z2 area, and label distribution of training samples from Dataset 2.

**Table 2.** Sample selection statistics of acoustic backscatter images.

| | Image Size | Class | 1 | 2 | 3 | 4 | 5 | 6 |
|---|---|---|---|---|---|---|---|---|
| Z1 area [1] | 492 × 505 | Train sample | 534 | 506 | 599 | 606 | 183 | 369 |
| | | Test sample | 811 | 571 | 634 | 677 | 270 | 240 |
| Z2 area [2] | 717 × 634 | Train sample | 419 | 368 | 342 | 325 | 442 | 383 |
| | | Test sample | 463 | 405 | 370 | 283 | 428 | 333 |

[1] The class numbers represent the sediment category, which are fine silt, median silt, fine sand, median sand, granule, and undetected area. [2] The sediment categories of the Z2 area are: silt-clay-sand, sand-silt-clay, silty-clayey sand, sand, gravel sand, and gravelly sand.

Dataset 2 was the result of a coastal survey conducted by the National Oceanic and Atmospheric Administration (NOAA), Stony Creek University, and Rhode Island University from 2001 to 2013 in the Long Island sea area. Its regional location is shown in Figure 6b. The backscattered data collected during this period were stitched into a 1 × 1 m results map. We selected the area near the coastline and marked it as Z2. The refined classification provides basic data for the study of biological population distribution and sediment types. Referring to the classification results for a large-scale object-oriented image [45], the sedimentary types in the Z2 area were: silt–clay–sand, sand-silt-clay, silty-clayey-sand,

sand, gravel sand, and gravelly sand (Figure 6b). The number of training and testing samples are shown in the last two rows of Table 2.

Some parameters of the proposed method were set as follows. The image neighborhood window size was set to 9. The SDAE network includes two hidden layers (L1 = 50, L2 = 40). The learning rate was $10^{-3}$, a certain proportion of the zero masked fraction was set as 0.1, and the epochs and batch size were 100 and 1000, respectively. In regard to the family model of ELM, the following settings were used: the intermediate hidden layer $H = 50$, the regularization coefficient $C = 0.5$, the Parzen window ratio $h = 5$, and the weight adjustment factor $\eta = 4$. In addition, the maximum iteration times *itmax* and population number *popn* in the PSO and GA algorithm are 100 and 40, respectively. Here, the inertia weight range [*wmin*, *wmax*] = [0.4, 0.8], the population acceleration factor *c*1 and the global acceleration factor *c*2 were both 0.2, and the position range [*a*, *b*] and velocity range [*m*, *n*] were [−1, 1]. The crossover probability *pc* and mutation probability *pm* in GA were 0.9 and 0.1, respectively.

### 3.2. Results of Feature Extraction

In order to verify the reliability of the SDAE used in extracting the deep-seated feature information from the pixels, the Gabor filter [12], CNN [13] and SAE [14] approaches are used for comparison. The Gabor filter extracts texture features from backscatter image in 6 directions and 4 scales, and uses a Gaussian smoothing filter for post-processing. The parameters of the CNN were set as follows: the size of the neighborhood window was 9, and convolution and pooling operations were performed at a scale of 28. The number of iterations was 100 and the learning rate was 0.1. The number of convolution layers was 2. There were 6 and 12 output maps, and the kernel size was 5. The scale in the subsampling layer was 2. The SAE had the same parameters as the SDAE, except for the zero masked fraction. The MELM classifier served as the classification basis, and Z1 and Z2 areas were used for experiments. The performances of the feature extraction techniques were analyzed according to the intensity of the noise interference, testing samples evaluation indexes and the prediction effect of the full map. Experimental results are shown in Table 3 and Figures 7 and 8.

**Table 3.** Testing set evaluation indexes of feature extraction methods in the Z1 and Z2 areas.

|  |  | $CA_1$ | $CA_2$ | $CA_3$ | $CA_4$ | $CA_5$ | $CA_6$ | *OA* | *Kappa* | *RMSE* |
|---|---|---|---|---|---|---|---|---|---|---|
| | Gabor | 64.7% | 60.7% | 90.0% | 92.7% | 78.7% | 97.7% | 78.7% | 0.733 | 0.4360 |
| Z1 area | CNN | 100% | 70.0% | 100% | 96.9% | 56.9% | 93.1% | 85.2% | 0.812 | 0.3832 |
| | SAE | 98.1% | 77.1% | 85.3% | 77.1% | 73.8% | 94.6% | 84.4% | 0.806 | 0.3256 |
| | SDAE | 99.7% | 76.6% | 99.3% | 88.8% | 71.9% | 96.8% | 89.3% | 0.868 | 0.3110 |
| | Gabor | 100% | 77.4% | 93.1% | 100% | 100% | 87.1% | 91.9% | 0.902 | 0.2888 |
| Z2 area | CNN | 100% | 75.7% | 94.3% | 99.6% | 100% | 95.9% | 93.0% | 0.915 | 0.2765 |
| | SAE | 100% | 84.8% | 86.6% | 100% | 99.8% | 86.7% | 92.8% | 0.913 | 0.2637 |
| | SDAE | 100% | 95.4% | 81.0% | 97.6% | 100% | 100% | 95.4% | 0.944 | 0.2601 |

$CA_i$ ($i = 1, \ldots, 6$) represents the category accuracy of each sediment.

Table 3 shows that the overall indexes of the Z2 test set are better than those of Z1, which reflects that residual errors decrease the quality of sediment classification. In particular, the Gabor filter extracts the texture features of the surface layer, which is susceptible to the influence of residuals, resulting in poor overall classification results. In regard to the category accuracy (*CA*), many *CA* values from the SDAE are superior to other approaches. The $CA_2$ value of the SDAE for sand-silt-clay can reach 95.4% in the Z2 testing samples, which is about 10% higher than reached with other methods. In addition, the *OA*, *kappa* and *RMSE* values of the SDAE are improved by at least 4%, 0.05, 0.014 and 2%, 0.03, 0.003, respectively. In general, more indicators demonstrate that feature values extracted by the SDAE are more meaningful and universal for comprehensive classification.

To further compare the feature extraction performance of the four approaches, the sediment prediction maps of the entire area were applied to characterize their overall generalization universality. Figure 7a shows that the Gabor filter easily confuses fine silt and medium silt, causing a large area of

misjudgment. Many adjacent sediment types interfere with each other in Figure 8a, such as between gravelly sand and sand-silt-clay. The sediment boundary line treated by the CNN was disturbed in the form of a zigzag (Figures 7b and 8b). Figures 7c and 8c display the classification problem of zigzag or scatter distribution, reflecting that the SAE inevitably brings the residual errors from the original image into the feature vector, thus providing error information for the type judgment of subsequent classifiers. However, Figures 7d and 8d processed by the SDAE show that the classification of sediment is clearly hierarchical and continuous, except for a few scattered points. SDAE introduces the zero masked fraction into the image so that the training network is able to learn how to remove noise automatically. The feature information extracted by the SDAE is more robust, reducing the interference of residual error on the subsequent classification.

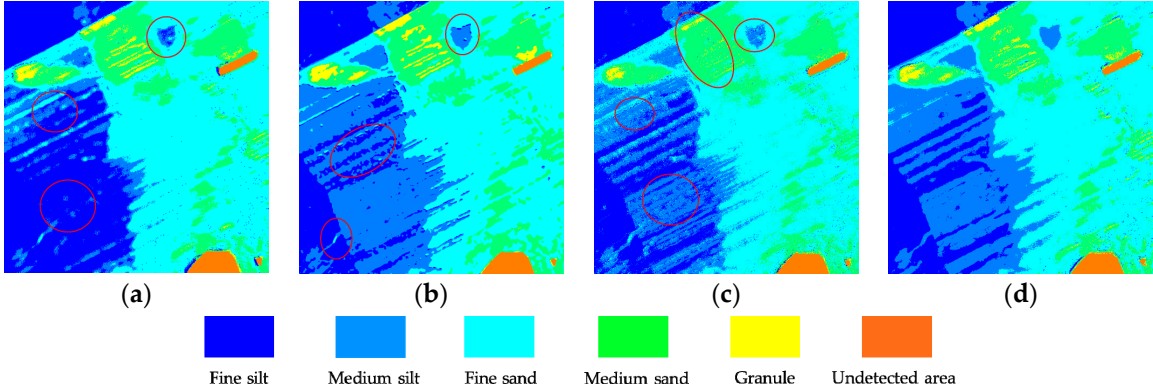

**Figure 7.** Image classification results of various feature extraction approaches in the Z1 area: (**a**) Gabor, (**b**) convolutional neural network (CNN), (**c**) stacked auto encoder (SAE), and (**d**) SDAE.

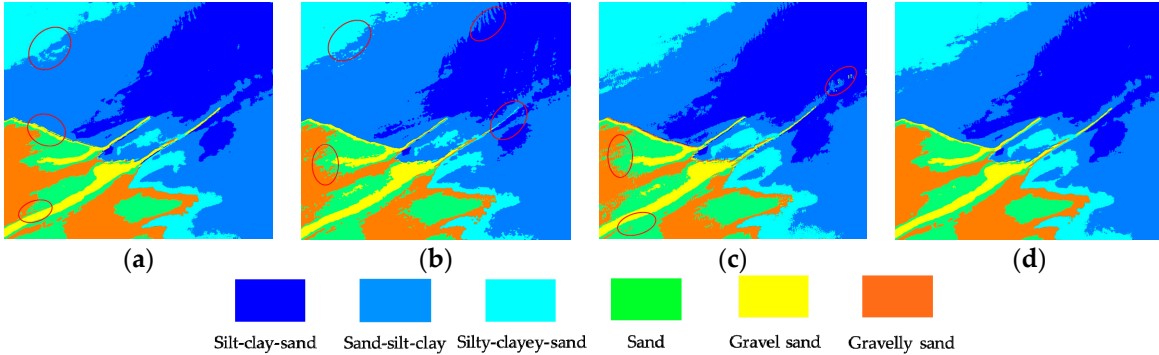

**Figure 8.** Image classification results of various feature extraction approaches in the Z2 area: (**a**) Gabor, (**b**) CNN, (**c**) SAE, and (**d**) SDAE.

### 3.3. Results of Classifiers Design

In order to show that the MELM classifier can effectively achieve the classification of sediment features, the performance of other classifiers are used for comparison. The selected classifiers were: (1) RF; (2) SVM; (3) GA-SVM; (4) PSO-SVM; (5) ELM; (6) RELM; (7) GA-RELM; and (8) our MELM. Among them, *ntree* = 50 was the number of decision trees in the RF. The SVM family used cross-validation to optimize the model. The deep-level features extracted by the SDAE served as the input vectors of every classifier. Images in Z1 and Z2 areas were used for the experiment, and the classification performance of each classifier was measured according to the objective indexes of testing samples and the subjective visual effects of the entire image. The experimental results are shown in Tables 4 and 5 and Figures 9 and 10.

Table 4 shows that the classification quality of the RF for fine silt and granule is low. The SVM families are not very sensitive to the classification of medium silt and undetected areas. In addition,

the ELM achieves more stable and balanced accuracy for various categories related to the basic SVM. Compared with the ELM classifier, the RELM improves the robustness of the training samples, while the GA-RELM also weakens the random interference of network parameters with the model. Better still, the *OA*, *Kappa*, and *RMSE* values of the MELM classifier are 89.3%, 0.868, and 0.311, which are at least 1.8%, 0.02, and 0.11 better than the other methods, respectively. This result reveals that our MELM model can improve the quality of classification under the condition of introducing incorrect training samples.

**Table 4.** The test set evaluation indexes of each classifier in the Z1 area.

|  | $CA_1$ | $CA_2$ | $CA_3$ | $CA_4$ | $CA_5$ | $CA_6$ | *OA* | *Kappa* | *RMSE* |
|---|---|---|---|---|---|---|---|---|---|
| RF | 89.6% | 82.8% | 99.8% | 87.7% | 64.7% | 95.7% | 87.0% | 0.840 | 0.4505 |
| SVM | 94.4% | 73.7% | 96.8% | 95.7% | 74.2% | 63.5% | 85.5% | 0.822 | 0.4585 |
| GA-SVM | 95.4% | 72.1% | 100% | 95.8% | 71.4% | 81.3% | 87.3% | 0.843 | 0.4135 |
| PSO-SVM | 95.5% | 71.7% | 100% | 94.6% | 72.6% | 78.0% | 86.9% | 0.839 | 0.4292 |
| ELM | 91.0% | 73.0% | 99.4% | 84.6% | 88.2% | 95.6% | 86.8% | 0.836 | 0.4521 |
| RELM | 86.3% | 75.7% | 98.8% | 85.7% | 95.5% | 93.2% | 87.3% | 0.842 | 0.4376 |
| GA-RELM | 97.4% | 77.4% | 90.4% | 82.6% | 84.4% | 99.0% | 87.5% | 0.845 | 0.4293 |
| Our MELM | 99.7% | 76.6% | 99.3% | 88.8% | 71.9% | 96.8% | 89.3% | 0.868 | 0.3110 |

**Table 5.** The test set evaluation indexes of each classifier in the Z2 area.

|  | $CA_1$ | $CA_2$ | $CA_3$ | $CA_4$ | $CA_5$ | $CA_6$ | *OA* | *Kappa* | *RMSE* |
|---|---|---|---|---|---|---|---|---|---|
| RF | 100% | 75.7% | 98.0% | 100% | 99.3% | 81.5% | 90.7% | 0.888 | 0.3290 |
| SVM | 100% | 73.5% | 90.3% | 100% | 99.1% | 69.4% | 86.4% | 0.836 | 0.3826 |
| GA-SVM | 100% | 75.8% | 95.3% | 100% | 97.5% | 73.5% | 88.3% | 0.859 | 0.3497 |
| PSO-SVM | 100% | 75.8% | 95.0% | 100% | 98.4% | 73.2% | 88.4% | 0.860 | 0.3322 |
| ELM | 100% | 78.8% | 94.9% | 100% | 100% | 89.8% | 93.1% | 0.916 | 0.2878 |
| RELM | 100% | 77.4% | 97.3% | 99.6% | 99.3% | 94.5% | 93.5% | 0.922 | 0.2837 |
| GA-RELM | 100% | 79.4% | 97.1% | 100% | 100% | 90.6% | 93.6% | 0.923 | 0.2615 |
| Our MELM | 100% | 95.4% | 81.0% | 97.6% | 100% | 100% | 95.4% | 0.944 | 0.2601 |

There are many large misjudgment areas and scatter confusion problems in Figure 9a, which indicate that the RF is unable to deal with the interference of residual error on sediment type. Figure 9b–d shows that the two kinds of sediments with similar intensity values make it difficult for the SVM, GA-SVM and PSO-SVM models to weaken their mixed classification interference, such as for granule and undetected area. In addition, Figure 9e shows that the ELM has a problem of inaccurate prediction for the unbalanced sediments of granule. Figure 9f–h shows that the classification interference of fine silt to medium silt can be improved, especially with the proposed MELM model. Better still, the classification quality of the original strip shadow is improved to a great extent, such as in the middle image where the medium silt acts on fine sand. Only in the lower right corner of Figure 9h does a small part of scattered point distribution exist.

Table 5 shows that the distribution of sediment types in the Z2 area is more regular and distinct, which makes the overall classification accuracy of this area generally higher. The overall accuracy of the RF test set is at the middle level. The *OA*, *kappa* and *RMSE* values of the SVM are the smallest, at only 86.4% 0.836 and 0.3826, respectively, as the SVM family is not very sensitive to the classification of gravel sand. In addition, the ELM family is superior to the SVM family in terms of classification quality and overall classification accuracy. In particular, the MELM model is far ahead of other techniques for sand-silt-clay sediments ($CA_2$ value), with an improvement of at least 16%. In summary, many indexes reflect the superior performance and application potential of the MELM classifier. The *OA*, *kappa* and *RMSE* values of our MELM are increased by 2.3%, 0.028 and 0.0277, respectively, compared with the basic ELM model.

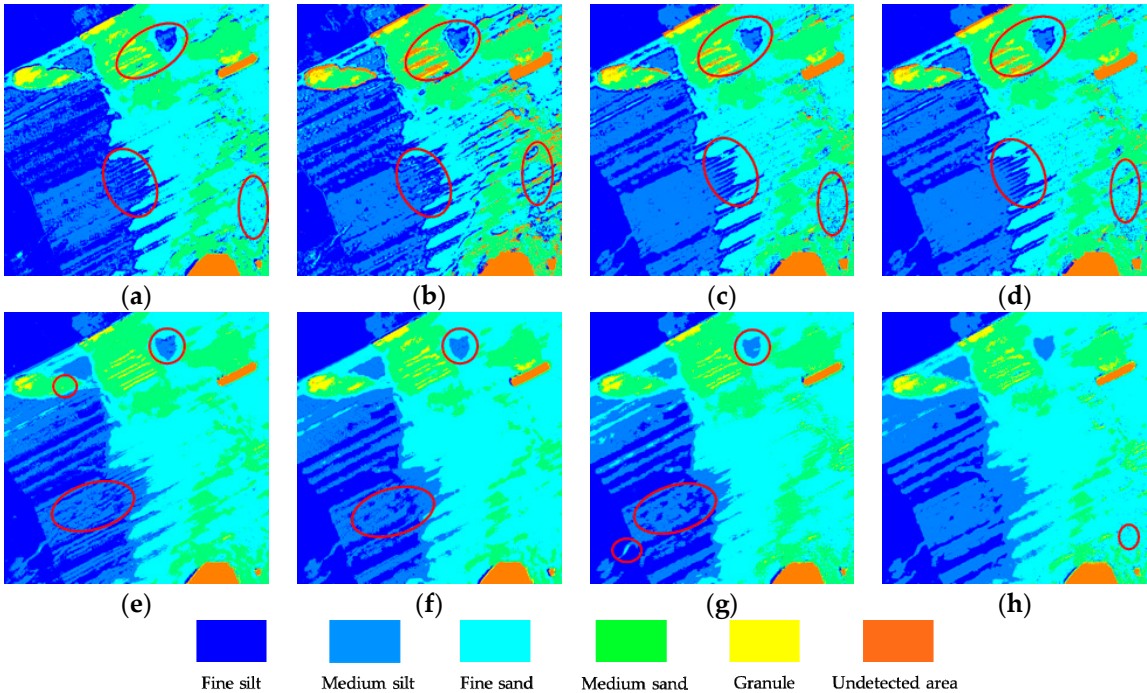

**Figure 9.** Image classification results of various classifiers on the Z1 area: (**a**) random forest (RF), (**b**) support vector machine (SVM), (**c**) GA-SVM, (**d**) PSO-SVM, (**e**) ELM, (**f**) RELM, (**g**) GA-RELM, and (**h**) our MELM. Several areas are marked to show the classification performance.

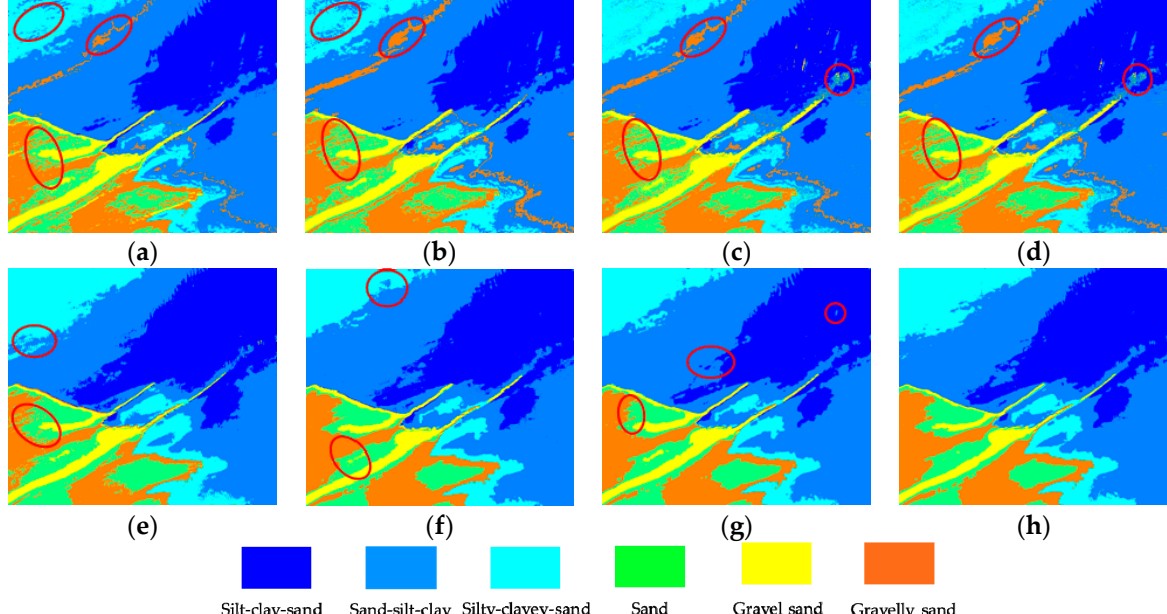

**Figure 10.** Image classification results of various classifiers on the Z2 area: (**a**) RF, (**b**) SVM, (**c**) GA-SVM, (**d**) PSO-SVM, (**e**) ELM, (**f**) RELM, (**g**) GA-RELM, and (**h**) our MELM. Several areas are marked to show the classification performance.

In regard to the prediction map of the Z2 area, Figure 10a–d shows that the wake sediment flow of the gravelly sand seriously affects the overall prediction effect, and that this kind of sediment also interferes with the classification quality of sand in the jagged form. Figure 10b shows that the SVM produces a large area of sand-silt-clay that interferes with the classification of silty-clayey-sand, while the GA-SVM and PSO-SVM only weaken this part of the problem (Figure 10c,d). Compared with

the ELM, the RELM and GA-RELM are able to weaken the jagged border and disperse the spots appropriately, however, the interference effect of gravel sand on sand is still powerless and insoluble. Figure 10h shows that the boundary of gravely sand sediment remains continuous.

## 4. Discussion

### 4.1. Comparison of Feature Extraction

Many residual errors and smear phenomena still exist in the acoustic backscatter image after pre-processing, requiring the feature extraction technology to have strong robustness. For the techniques based on image textures and statistical features, such as GLCM [9] and Gabor [12], the feature vectors are easily contaminated, and the results for sediment classification are not very satisfactory, as shown in Table 3 and Figure 7a. The CNN [13] generally relies on a certain amount of manually labeled sample data and is limited by the receptive field area and convolution kernel, meaning long-distance feature information is usually ignored. In addition, although SAE [14] is able to extract the deep feature information from the sediment image, it is also susceptible to the residual error of the original image, providing error information for subsequent classifiers when judging sediment types (Figure 7c). However, the SDAE introduces a certain zero masked fraction to enable the training network to adaptively learn to remove noise. In this way, the extracted feature information is more robust, reducing the interference of residual error with the subsequent feature classification.

### 4.2. Comparison between Classifiers

For fine sediment classification, the residual errors in the backscatter image also interfere with the selection of the training samples, inevitably bringing the signals with incorrect label into the network and affecting the classification performance. Among the various approaches, the RF method [22] is easily affected by the extracted sediment characteristics, and its decision number, splitting attributes and other key parameters will affect its efficiency and accuracy. When the intensity values are very close to each other, the SVM [23] produces a large area of misjudgment and pseudo flow phenomenon (Figures 9b and 10b). It is difficult for the GA–SVM and PSO–SVM models [25] to mitigate these problems; rather they optimize the randomness of their parameters to reduce the discrete distribution of spots.

Compared with the above classifiers, although the ELM [36] performs well in terms of generalization and operational efficiency, its robustness and randomness need to be improved. The RELM can weaken the weight of mislabeled samples to reduce the interference effect on the training network, and can balance the number of minority sediment categories. In regard to solving the randomness of the RELM network parameters, many objective indexes show that the MELM is better than the GA-RELM. In addition, the overall effects show that the prediction map of our MELM is more accurate and closer to the natural phenomenon in maintaining the sediment boundaries and in the classification of regional sediments. The liquid sand wave in the upper right corner of Figure 10h means it is difficult to make the category of silt-clay-sand change with the flow.

### 4.3. Ablation Study between ELM Families

In this section, ablation studies were carried out by fusing robust module and optimization components to demonstrate the effectiveness of our MELM classifier. The category accuracy (CA) and overall metrics were used for quantitative and qualitative evaluation. Tables 4 and 5 show that the RELM model has higher prediction accuracy for multiple categories compared with the ELM model, revealing that the RELM model is able to effectively improve the network robustness when residual errors exist in the training samples. In addition, in regard to the intelligent optimization of the GA-RELM model and our MELM model, the MELM model has a higher score for the classification of other sediments, except for gravel sand. For example, the category accuracy ($CA_1$, $CA_4$, $CA_6$) values for silt-clay-sand, sand, and gravel sand are 99.7%, 88.8% and 96.8%, respectively. Therefore, our

MELM model obtains more reliable scores and has an advantage in terms of multiple class evaluation indicators by integrating the functions of each module.

Figures 11 and 12 display the results for the overall evaluation indexes. Since the backscatter image in the Z2 area contains relatively fewer residual errors and noises, the values of the *OA*, *kappa* and *RMSE* are higher in the Z2 area. This also shows that the quality of the original data has greatly improved the accuracy of classification. Compared with the previous four classification methods, the overall metrics of the ELM family model can match the others or achieve higher scores in the Z1 and Z2 areas. In addition, the ablation module of the RELM provides accuracy improvement for the values of *OA*, *kappa* and *RMSE*. In particular, the PSO module in our MELM model undoubtedly achieves the best performance in the process of intelligent optimization, while the RMSE error values for label prediction reach 0.3110 and 0.2601 on the testing samples of Z1 and Z2 areas, respectively. Therefore, the combination of two ablation modules effectively enhances the test accuracy and the stability.

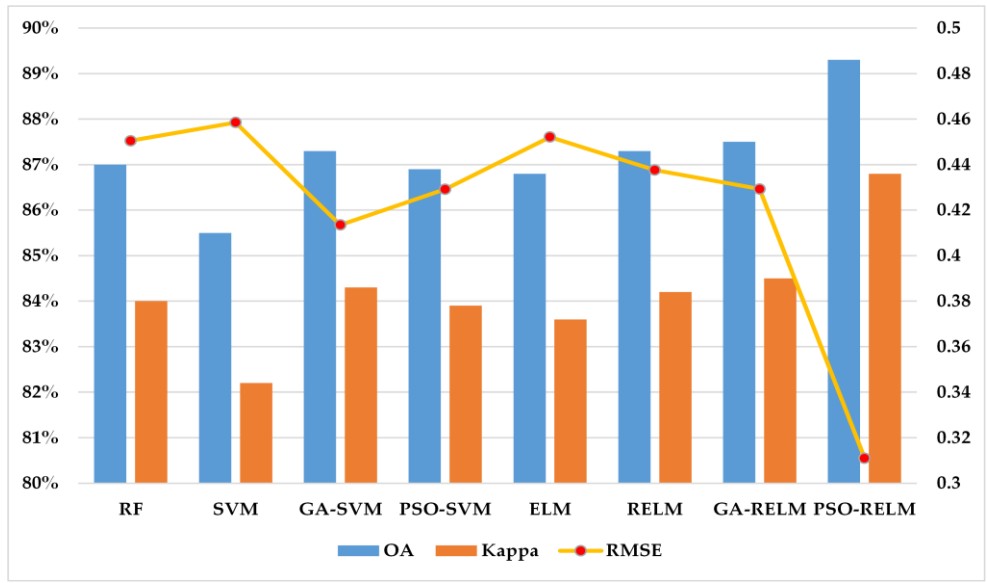

**Figure 11.** Overall evaluation metrics of the testing samples in the Z1 area.

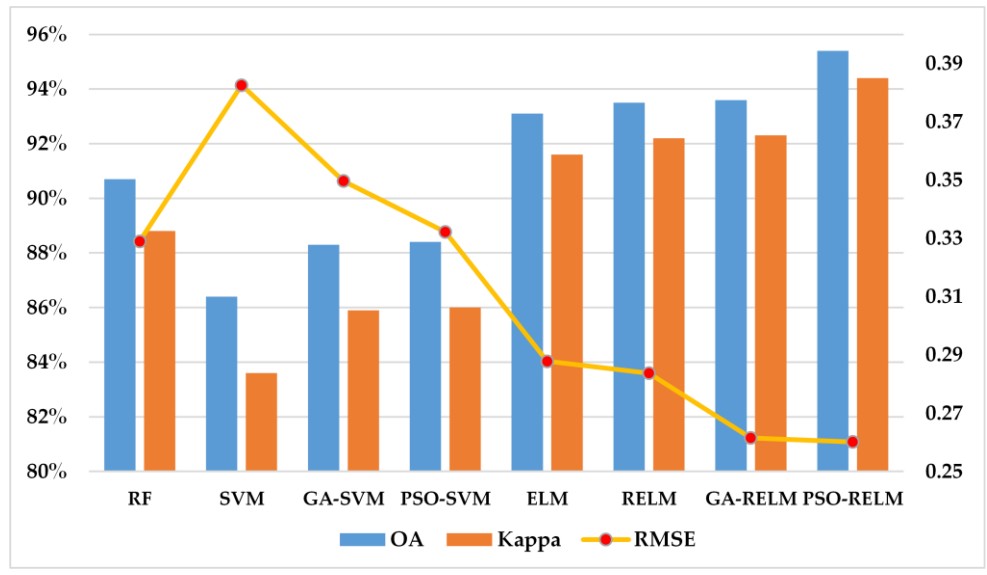

**Figure 12.** Overall evaluation metrics of the testing samples in the Z2 area.

## 5. Conclusions

This paper presents a sediment classification technique by uniting the SDAE and MELM models based on the acoustic backscatter image. In order to reflect the rationality and robustness of feature extraction, the Gabor, CNN, and SAE approaches are used for comparative analysis. The experimental results reveal that the SDAE introduces a certain zero masked fraction, so that the training network automatically learns how to remove noise and residual errors from the backscatter image. The feature information extracted in this way has a stronger representation, which weakens the interference of residual error on the subsequent feature classification. On the other hand, the performance of the optimal MELM classifier is demonstrated by comparison with seven other models. The experimental results indicate that the MELM model is able to effectively weaken the influence of the training sample with error information on the model network. In addition, the process of intelligent optimization reduces the randomness of the input parameters, improving the stability and robustness of the training network. In summary, the proposed SDAE-MELM technology improves the classification accuracy and stability, and the boundary lines of sediment types in the classification map are more continuous and conform to natural characteristics. In the future, in order to effectively achieve sediment classification in acoustic backscatter images with residual errors, certain post-processing methods will be adopted to eliminate the discrete spots that occur in the prediction map.

**Author Contributions:** P.Z. and G.C. conceived the model methodology. M.W. helped to build the paper framework. J.C. collected the data. P.Z. and M.W. wrote the initial draft. J.C. and Y.L. revised the manuscript. All authors have read and agreed to the published version of the manuscript.

**Funding:** The project was supported by the National Natural Science Foundation of China under Grant No.41674015, 41901296.

**Acknowledgments:** Dataset 1 from the USGS platform was derived from scientific research on the shallow geology, seabed structure, and geographical area of Bald Eagle Bay, Massachusetts. Dataset 2 was derived from a backscatter image from Long Island Coastal Survey in the MGDS platform. We are very grateful for the contributions of data collection and data pre-processing, and for the platform providers to the relevant data.

**Conflicts of Interest:** The authors declare no conflict of interest.

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
