# Peer review of "Sediment Classification of Acoustic Backscatter Image Based on Stacked Denoising Autoencoder and Modified Extreme Learning Machine"

_remotesensing, doi:10.3390/rs12223762_

Round 1

Reviewer 1 Report

Please see the attached PDF-file.

Author Response

Reviewer 1:

(1) Could the authors include the following references [1-2] dealing with neural networks for sediment classification? 

Response: It's our fault to ignore the two provided literatures. At your reminder, we have read them and have also added them to the introduction in the new version. The relevant contents have been modified to conform to the theme of the whole paragraph. For examples,

The original text is as follows: "Commonly unsupervised classification techniques include cluster analysis [18] and iterative self-organizing data analysis technology (ISODATA) [19]. These methods had low classification accuracy and only draw preliminary seabed sediment maps. Besides, the supervised classification techniques of seabed sediment contain neural network [20], multilayer perceptron [21], random forest (RF) [22] and support vector machine (SVM) [23]. However, the scalability, sensitivity of key parameters and robustness of these supervised classifiers were usually restricted. …"

It is amended as follows: "Commonly unsupervised classification techniques include K-means [18], self-organizing feature maps [19] and iterative self-organizing data analysis (ISODATA) [20]. These methods are greatly influenced by noise data, resulting in the rapid reduction of classification accuracy. Some post-processing methods are needed to optimize the classification effect when drawing the preliminary seabed sediment map, such as Bayesian technology [21]. Additionally, the supervised classification techniques of seabed sediments include neural network (NN) [22], multilayer perceptron [23], random forest [24] and support vector machine [25] approaches. However, the sensitivity of the key parameters and the robustness of these supervised classifiers are usually restricted. Ojha and Maiti applied Bayesian and NN to distinguish sediment boundaries in the Bering Sea slope area [26]. This method effectively improves the robustness of the algorithm when there is red noise in the data. Meanwhile, …"

[18] Cooper, K.M.; Bolam, S.G.; Downie, A.L.; Barry, J. Biological-based habitat classification approaches promote cost-efficient monitoring: An example using seabed assemblages. J. Appl. Ecol. 2019, 56, 1085-1098.

[19] Chakraborty, B.; Kaustubha, R.; Hegde, A.; Pereira, A. Acoustic seafloor sediment classification using self-organizing feature maps. IEEE Trans. Geosci. Remote Sensing. 2001, 39, 2722-2725.

[21] Karmakar, M.; Maiti, S.; Singh, A.; Ojha, M.; Maity, Bhabani, S. Mapping of rock types using a joint approach by combining the multivariate statistics, self-organizing map and Bayesian neural networks: an example from IODP 323 site. Mar. Geophys. Res. 2018, 39, 407-419.

[26] Ojha, M.; Maiti, S. Sediment classification using neural networks: an example from the site-U1344A of IODP Expedition 323 in the Bering Sea. Deep-Sea Res. II. 2013, 125, 202-213. http://dx.doi.org/ 10.1016/i.dsr2.2013.03.024.2.

(2) Are the colored points shown in figure 6a and b samples from the seafloor, where the sediment types have been determined by mechanical or chemical analysis? Please provide additional information about the training data.

Response: Thank you for your comments. The colored points shown in Figure 6a and b are the training samples that we manually select based on the backscatter intensity information. In addition, we also insist on referring to object-oriented image classification results of the original project. Their classification results are in view of actual seabed samples and sediment particle analysis indicators [1-2]. However, the processing results are for large scales, and a lot of post-processing work is required to form relatively considerable visual effects, such as manual removing noise spots. We found that their large-scale classification results are not very accurate for the sediment boundary. Therefore, this paper selects the local area to study the refined sediment classification, which is based on the acoustic backscatter image provided by the original project. Simultaneously, we found that there is a small amount of wrong sample information in the training samples we manually selected. In addition, there are residual errors and noise after the original image is preprocessed. Therefore, we use the unsupervised SDAE method for denoising feature extraction. The proposed MELM model is used to deal with the error sample information. Meanwhile, PSO is used to optimize the random parameter representation in the training network. Finally, we firmly believe that a high-precision and refined sediment classification can be achieved by referring to the results of large-scale classification and solving the problem of self-selection.

[1] Foster, D.S.; Baldwin, W.E.; Barnhardt, W.A.; Schwab, W.C.; Ackerman, S.D.; Andrews, B.D.; Pendleton, E.A.; Shallow geology, sea-floor texture, and physiographic zones of Buzzards Bay, Massachusetts: U.S. Geological Survey Open-File Report 2014–1220, 2016, https://dx.doi.org/10.3133/ ofr20141220.

[2] Zajac, R.N.; Stefaniak, L.M.; Babb, I.; Conroy, C.W.; Penna, S.; Chadi, D.; Auster, P.J. An integrated seafloor habitat map to inform marine spatial planning and management: a case study from Long Island Sound (Northwest Atlantic). In Seafloor Geomorphology as Benthic Habitat, 2nd ed.; Peter, T. Harris.; Elaine, Baker.; Eds.; Publisher: Elsevier, Netherlands, 2020, pp.199-217.

(3) From a practical perspective: How are the performance metrics influenced by the presence of organic matter (plants, rotten wood)? 

Response: This is a good thinking idea, involving the knowledge of interdisciplinary integration. There are a lot of plants, algae and mollusks growing in lakes or seabed at a certain depth. After the death of these organisms, they accumulate and decompose in the benthic environment and form organic sediment together with clay and silt. The organic matter in sediments is usually analyzed by studying the chemical composition of sediments to show their main sources of the difference and distinguish them. Since the acoustic backscatter image in this paper is based on the reflection signal strength of surface features to achieve the classification of sediments. Theoretically speaking, it is a fine classification process from the perspective of acoustic and physical characteristics of seabed features, combining with image processing technology. In the future, it is of great significance to understand the benthic environment by combining the knowledge of geophysics, geochemistry and biological characteristics.

(4) Is the classifier capable of differentiating between solid rock mineral sediments, and organic sediments? 

Response: It is a good suggestion. In order to show that the MELM classifier proposed in this article can distinguish between rocks and mines. We use a public data set to experiment. The sonar data set comes from UCI Machine Learning Repository, which predicts that the target object is mines or rocks based on the intensity values returned by a given sonar from different angles. There are a total of 208 observation samples, 60 input feature values and 1 output label [1]. The ratio of training samples to test samples is 2:1. This experiment has been run 10 times to show the superiority of the proposed classifier by its average test accuracy, as shown in Table 1. Table 1 displays that our MELM model can achieve higher prediction accuracy and more stability, and the average test accuracy can reach 90.28%. Therefore, we believe that the MELM classifier can distinguish the corresponding objects according to the attribute characteristics of related substances.

[1] Gorman, R.P.; Sejnowski, T.J. Analysis of hidden units in a layered network trained to classify sonar targets. Neural Netw. 1988, 1, 75-89.

Table 1. The prediction accuracy of each classifier in the sonar dataset

RF

GA-SVM

PSO-SVM

ELM

GA-RELM

Our MELM

Round 1

82.41%

83.33%

81.48%

78.70%

81.48%

87.04%

Round 2

75.93%

87.96%

77.78%

76.85%

79.63%

89.81%

Round 3

80.56%

83.33%

79.63%

75.93%

82.41%

91.67%

Round 4

79.63%

82.41%

82.41%

72.22%

86.11%

92.59%

Round 5

74.07%

79.63%

82.41%

78.70%

87.96%

87.96%

Round 6

68.52%

79.63%

82.41%

75.93%

81.48%

93.52%

Round 7

67.59%

78.70%

74.07%

79.63%

87.96%

92.59%

Round 8

80.56%

83.33%

74.07%

67.59%

80.56%

89.81%

Round 9

75.93%

87.96%

86.11%

75%

83%

90.74%

Round 10

77.78%

81.48%

86.11%

79.63%

83.33%

87.04%

Mean

76.3%

82.78%

80.65%

76.02%

83.43%

90.28%

(5) How are the model predictions affected by environmental conditions, e.g. water depth, water turbidity (floating particles), frequency of the signal transceiver, and velocity of the sonar platform/boat? In principle, the information extraction by machine learning models from remote sensing data is prone to ambient conditions, which can significantly reduce the quality of prediction (i.e. lower performance metrics).

Response: We agree with this opinion. Environmental factors usually affect the prediction accuracy of classification models. We intend to explain the impact of different factors on the classification model from the following aspects.

1) The water turbidity and the speed of the sonar platform/boat interfere with the collection of reflected signals, which is not conducive to the formation of sonar image units. Generally, this kind of influence factor will be corrected in the preprocessing stage of CARIS or HYPACK software. If these factors are not extreme, the correction effect is satisfactory, and the resulting sonar image noise and line problems can be effectively suppressed. Therefore, it is necessary to improve the quality of the original image for subsequent feature extraction and classification model prediction.

2) Water depth can provide a lot of characteristic information such as slope, direction, curvature, and terrain variability. In the past, GLCM and Gabor filtering were applied to extract their features. Then, the extracted feature parameters were used as the input of the subsequent classification model. However, the preprocessing image of water depth contains residual errors and noises, which is the same as acoustic backscatter image. This problem makes the extracted feature vector contaminated, which reduces the classification performance of subsequent classifiers to a certain extent. For example, the test accuracy of the backscatter image is only 78.7% in the Z1 area of this article. Therefore, the quality of water depth indirectly affects the classification accuracy of the prediction model. However, the original project did not provide the information on public water depth data. Partial data catalog for dataset 1 is shown in Figure 1 [1]. Fortunately, the acoustic backscatter image can be downloaded in the original project. We can also work on the study of refined sediment classification based on the relationship between sonar echo intensity and seafloor features.

3) According to different frequency signal transmitters, the relationship between the echo intensity and the attribute categories of seabed features is studied. The red box in Figure 2 shows the classification result of K-means [2]. The echo intensity can effectively distinguish the types of ground objects under different beam incidence angles. However, the items in the experimental data set are all acoustic backscatter data collected at a certain frequency. Therefore, we can only classify sediment from the perspective of image classification and the reflected signal strength of the sediment. In the future, we firmly believe that it will be possible to study the influence of total echo intensity at multiple frequencies on the classification model, and it may be necessary to perform fuzzy voting to select which model or frequency is better.

[1] Foster, D.S.; Baldwin, W.E.; Barnhardt, W.A.; Schwab, W.C.; Ackerman, S.D.; Andrews, B.D.; Pendleton, E.A.; Shallow geology, sea-floor texture, and physiographic zones of Buzzards Bay, Massachusetts: U.S. Geological Survey Open-File Report 2014–1220, 2016, https://dx.doi.org/10.3133/ ofr20141220.

[2] Zhao, J.; Yan, J.; Zhang H.; Meng, J. A New Radiometric Correction Method for Side-Scan Sonar Images in Consideration of Seabed Sediment Variation. Remote Sens-Basel. 2017, 9, 575.

Figure 1. Partial data catalog for dataset 1

Figure 2. Relationship between echo intensity and feature attributes.

(6) Some words are improperly used or misspelled. Pay attention to the capital letters at the beginning of the sentence. Some grammar and sentences need to be rephrased.

Response: Thank you for your reminder and correction. We have revised the above-mentioned problems in the new version according to your prompts. In addition, other similar situations in the full text have also been corrected and improved. The following points need to be corrected are listed in the original lines.

On page 2 line 49: "personality" was revised as "property".

On page 2 line 92, On page 7 line 227, On page 9 line 272 and 280: The capital letters at the beginning of the sentence have been corrected.

On page 10 line 314: "sand-silent-clay" was corrected to "sand-silt-clay".

On page 2 line 77, On page 7 line 216, On page 15 line 446: Thank you for the revised content, which is conforms to the corresponding expression.

On page 11 line 345-348: the original sentence of "Figures 9b-d display that two kinds of sediments with similar intensity values make the SVM, GA-SVM and PSO-SVM models are difficult to weaken their mixed classification interference, such as granule and undetected area." was rephrased as " Figures 9b-d show that the two kinds of sediments with similar intensity values make it difficult for the SVM, GA-SVM and PSO-SVM models to weaken their mixed classification interference, such as granule and undetected area. ".

Finally, we have edited the work on language style and readability with the participation of native English speakers. We really hope that there has been a substantial improvement in the level of programming and language.

Figure 3. Editing proof of English language

(7) On page 8 table 1: Please describe the symbols in the table, which were not previously mentioned in the text.

Response: Sorry for our negligence. We have added the definition of their symbols in the new version. As a note, mathematical symbols of objective indicators are expressed on page 8 line 251-254. M is the number of categories. Numij represents the number of i-th sample predicted to be j-th class. Numii represents the number of i-th sample predicted to be i-th class. Numi is defined as the number of class i-th real samples. Numj represents the number of samples predicted in class j-th. Nc is the total number of test samples. yij and ycij are the position index value and prediction value of the test label, respectively.

(8) On page 12 line 365: Do the authors mean 2.3%?

Response: I'm sorry for our writing mistakes. In order to fully explain that it is a spelling error, we use the confusion matrix results to illustrate. Confusion matrix covers the data sources of multiple indicators, such as OA, Kappa coefficients and CA. Figure 4 shows some index coefficients of each classification model. Figure 4e and Figure 4h clearly show that the OA values are 93.1% and 95.4%, respectively. Compared with the ELM model, our MELM improves the overall prediction accuracy by 2.3%. In addition, Kappa coefficient can be obtained according to the index calculation formula in Table 1 of the original text.

(a)    RF

(b)    SVM

(c) GA-SVM

(d)    PSO-SVM

(e) ELM

(f) RELM

(g) GA-RELM

(h) our MELM

Figure 3. Confusion matrix of classification prediction models in Z2 area

(9) On page 12 table 5: Why is the CA value for the proposed MELM model lower than for the other models?

Response: Thank you for your comments. We compare the results of GA-RELM model (other models can also be used) to explain why the CA3 value of our MELM model is relatively low, as shown in Figure 5. The diagonal line in Figure 4 shows the predicted number of real samples. We select 370 samples for the silty-clayey-sand characterized by CA3 (see Table 2 in the original text). The red box in Figure 2b shows that the number of silty-clayey-sand predicted by our MELM model is 354, while GA-RELM can only predict 269. Since there are 73 false predictions with a relatively large proportion, the CA3 value of category prediction accuracy is lower than other models. In addition, silty-clayey-sand has more contact with other sediments, especially the backscattering intensity value of sand-silt-clay is closer, which causes the classification model to easily misclassify it. The misclassification results of other categories are all accumulated in this category. In general, the CA value of MELM model is lower than other models, but the other objective indexes of MELM model are optimal.

(a) GA-RELM

(b) Our MELM

Figure 4. Emphatic representation of confusion matrix

Reviewer 2 Report

Whilst I found the comparative description of techniques interesting, I found it hard to follow given the poor use of English.

It would be good to know which scheme gave the best results compared with ground truth

There was poor use of the definite and indefinite article (the, a etc). Sometimes I could guess what was meant, but sometimes I was in doubt

On line 345 there is a missing verb ("is"?)

I have insufficient experience with these techniques to judge the validity of the explanations

Author Response

(1) Whilst I found the comparative description of techniques interesting, I found it hard to follow given the poor use of English.

Response: We apologize for the poor language of our manuscript. The repeated addition and removal of sentences obviously led to poor readability. Therefore, we have now committed to language and readability work, and have modified the terminology throughout the text as appropriate. In addition, native English speakers have involved to edit the style of language. We really hope that the process and language level have been substantially improved.

(2) It would be good to know which scheme gave the best results compared with ground truth.

Response: We are sorry for our negligence, let's elaborate on our work here. It is known that the sediment classification of the acoustic backscatter image includes two main steps: feature extraction and classification prediction (classifier design). In this paper, the feature extraction of SDAE and the prediction model of MELM are used to realize the fine sediment classification. Three experimental schemes are designed to show the advantages of each module. 1) In feature extraction experiments, Gabor, CNN and SAE approaches are adopted to compare with SDAE method. The results and analysis are shown in Section 3.2 and 4.1. The results demonstrate that SDAE method enables the training network to adaptively learn to remove residual errors and noise from the backscatter image, which weakens the interference of residual error on subsequent feature classification. 2) Classification prediction is based on feature extraction, and RF, SVM, GA-SVM, PSO-SVM, ELM, RELM, GA-RELM are applied to compare with the proposed MELM model. The results and analysis are displayed in Section 3.3 and 4.2. The results indicate that the reliability and robustness of our MELM perform other approaches from the overall classification map and comprehensive indexes. 3) MELM model includes the theory of RELM and PSO process with optimal parameter selection. We performed ablation studies in 4.3 Section. The results show that MELM model inherits the robustness of RELM model and the randomness reduction of PSO intelligent optimization.

(3) There was poor use of the definite and indefinite article (the, a etc). And some grammatical problems exist in the article, such as On line 345 there is a missing verb ("is"?).

Response: We thank the reviewer for pointing out this issue. We have revised the articles, verbs, attributive clauses and adverbial clauses in the new version. In addition, we have involved native English speakers for language corrections. We sincerely hope that the revised article will meet your criteria.

Figure 1. Editing proof of English language
